# A Bacterial Form I’ Rubisco Has a Smaller Carbon Isotope Fractionation than Its Form I Counterpart

**DOI:** 10.3390/biom13040596

**Published:** 2023-03-26

**Authors:** Renée Z. Wang, Albert K. Liu, Douglas M. Banda, Woodward W. Fischer, Patrick M. Shih

**Affiliations:** 1Division of Geological and Planetary Sciences, Caltech, Pasadena, CA 91125, USA; 2Environmental Genomics and Systems Biology Division, Lawrence Berkeley National Laboratory, Berkeley, CA 94720, USA; 3Department of Plant and Microbial Biology, University of California, Berkeley, Berkeley, CA 94720, USA; 4Feedstocks Division, Joint BioEnergy Institute, Emeryville, CA 94608, USA; 5Innovative Genomics Institute, University of California, Berkeley, Berkeley, CA 94720, USA

**Keywords:** carbon isotope fractionation, cyanobacteria, rubisco, evolution

## Abstract

Form I rubiscos evolved in Cyanobacteria ≥ 2.5 billion years ago and are enzymatically unique due to the presence of small subunits (RbcS) capping both ends of an octameric large subunit (RbcL) rubisco assembly to form a hexadecameric (L_8_S_8_) holoenzyme. Although RbcS was previously thought to be integral to Form I rubisco stability, the recent discovery of a closely related sister clade of octameric rubiscos (Form I’; L_8_) demonstrates that the L_8_ complex can assemble without small subunits (Banda et al. 2020). Rubisco also displays a kinetic isotope effect (KIE) where the 3PG product is depleted in ^13^C relative to ^12^C. In Cyanobacteria, only two Form I KIE measurements exist, making interpretation of bacterial carbon isotope data difficult. To aid comparison, we measured in vitro the KIEs of Form I’ (*Candidatus* Promineofilum breve) and Form I (*Synechococcus elongatus* PCC 6301) rubiscos and found the KIE to be smaller in the L_8_ rubisco (16.25 ± 1.36‰ vs. 22.42 ± 2.37‰, respectively). Therefore, while small subunits may not be necessary for protein stability, they may affect the KIE. Our findings may provide insight into the function of RbcS and allow more refined interpretation of environmental carbon isotope data.

## 1. Introduction

Rubisco (ribulose-1,5-bisphosphate carboxylase-oxygenase) is a keystone enzyme linking Earth’s inorganic and organic carbon cycles, which makes it a prime target for bioengineering associated with food systems and carbon sequestration. It is the most abundant protein on Earth today [1] because it catalyzes the essential carbon fixation step in one of the most ecologically dominant carbon-fixing metabolisms, the Calvin Benson Bassham (CBB) cycle in oxygenic photosynthesis [2]. Rubisco and oxygenic photosynthesis form the basis of our food web in terrestrial and marine systems because both eukaryotic and bacterial primary producers utilize rubisco to convert inorganic carbon (CO_2_ and HCO_3_^−^) into biomass that is then consumed by heterotrophs up the food chain. In addition, the annual flux of CO_2_ fixed by rubisco is very large, representing the single most massive flux in the global carbon cycle. Gross primary productivity (GPP), which accounts for all forms of carbon fixation but is vastly dominated by oxygenic photosynthesis, is ≈120 Gt C yr^−1^ in terrestrial [3] and ≈100 Gt C yr^−1^ in marine environments [1,4], compared to ≈10 Gt C yr^−1^ emitted of anthropogenic fossil CO_2_ [5]. Therefore, multiple efforts exist to engineer a ‘better’ rubisco that fixes more CO_2_ in order to increase crop yields and sequester anthropogenic CO_2_, among many other motivations (see [6] for review).

However, these bioengineering approaches are informed to a degree by our current understanding of rubisco’s evolutionary history, which itself is based on our understanding of past Earth environments. These evolutionary questions largely center on the canonical paradox that, despite being a central metabolism enzyme, rubisco is: (i) ‘slow,’ and (ii) ‘confused’ because it can fix O_2_ instead of CO_2_ [7], which invokes a salvage pathway that costs ATP, reducing power, and carbon [8]. This paradox is usually resolved by considering the atmospheric composition when rubisco first evolved more than 2.5 billion years ago, when CO_2_ was much higher (potentially up to ≈20x present atmospheric levels in the Precambrian [9]) and O_2_ only existed at trace levels [2]. However, in a Shakespearean tragedy, once rubisco was incorporated into the greater metabolism of oxygenic photosynthesis, it poisoned the very world it came from—successful CO_2_ fixation was coupled with oxygenation that permanently changed the atmosphere to one where O_2_ is dominant (≈20%) and CO_2_ is trace (≈0.04%). Now saddled with a rubisco evolved from a chemical world that no longer exists, diverse land plants, algae, and Cyanobacteria have independently evolved complex CO_2_ concentrating mechanisms (CCMs) that effectively hyper-concentrate CO_2_ at the expense of O_2_ around rubisco [10]—in effect, replicating the ancient atmosphere within their own cells. Those without CCMs (e.g., C3 plants) instead accommodate the low carboxylation rate by producing this enzyme at such high concentrations that up to 65% of all soluble protein in leaf extracts is just rubisco [11]. This narrative, contingent on our understanding of the geologic carbon cycle, suggests either that rubisco is an ‘accident’ of evolutionary history, or that it is truly the optimal enzyme designed by evolution for a difficult task. Therefore, a better understanding of the evolutionary history of this enzyme is useful for rubisco engineering efforts.

Rubisco is also notable because it displays a large carbon kinetic isotope effect (KIE) where it preferentially fixes ^12^CO_2_ over ^13^CO_2_ due to the rate of carboxylation being slightly faster for ^12^CO_2_ [12]. This effect is typically reported in delta (δ^13^C) and epsilon (ε) notation in units of per mille (‰), where δ^13^C = [^13^R_sa_/^13^R_ref_ − 1]*1000 and ^13^R is the ratio of ^13^C/^12^C in the sample or reference, respectively. ε is roughly the difference in δ^13^C between the product and the reactant (ε_Rubisco_ ≈ δ^13^C_3PG_ − δ^13^C_CO2_). Thirteen unique rubisco KIEs (ε_Rubisco_ values) have been measured across a limited range of phylogenies and species, but measurements so far indicate that rubisco fractionates at roughly 20–30‰ (for a recent review see [13]).

This KIE is useful because it allows one to track mass flux through complex systems in both modern and ancient environments [14], and because it may give insight into non-isotopic enzyme kinetics [15]. Since all biomass is ultimately synthesized from 3PG in autotrophs utilizing the CBB cycle, rubisco’s KIE is inherited by bulk biomass such that organic carbon is also relatively depleted in ^13^C relative to inorganic carbon. Therefore, when incorporated into larger metabolic models of carbon fixation, rubisco KIEs have facilitated the estimation of water use efficiency in plants [16], the efficiency of carbon fixation in bacterial and eukaryotic algae [17], the contribution of terrestrial plants to global GPP [18], and the proportion of C3 vs. C4 plants in mammalian diets [19], among many other examples. Similarly, in ancient environments, it has been used to estimate paleo atmospheric CO_2_ levels [20,21], track the inorganic and organic carbon cycle through time [22], and the diet of ancient mammals [23]. In addition, rubisco KIEs have been used to support interpretation of important non-isotopic kinetic parameters such as the inverse correlation between specificity for CO_2_ over O_2_ (S_C/O_) and rate of carboxylation (V_C_) [15]. Therefore, knowing the KIEs of many rubiscos is valuable because it facilitates empirical measurements of mass flux in many systems, natural and engineered, where other measurements may be difficult.

However, the landscape of rubisco evolution and its effect on KIE has not been well characterized. This is particularly true in Cyanobacteria, the organism within which rubisco and oxygenic photosynthesis is thought to have evolved. Most rubisco KIEs have been measured for Form IB rubiscos from plants, and in Cyanobacteria, only one Form IA and one Form IB rubisco KIE have been measured ([24,25], for a recent review see [13]). This is particularly important for reconstructing paleo pCO_2_ levels because direct measurements of the atmosphere from ice core records only extend back ≈1 million years [26], so for the remainder of Earth’s 4.567 billion year history we must rely on indirect measurements such as the carbon isotope record: globally assembled measurements of δ^13^C in the inorganic or organic carbon bearing phases of sedimentary rocks [27]. Interpretation of these records relies on geochemical models, largely based on extant modern organisms, that incorporate the rubisco KIE to explain most of the offset in δ^13^C between inorganic and organic carbon pools (see [28] for recent review of current models). These models inform our understanding of ancient atmospheres which in turn can influence our ideas of rubisco evolution in the past and engineering strategies in the present. It is therefore critical that we better understand the evolution of rubisco’s KIE through time because it underlies many assumptions we make when interpreting both the past and present. 

We therefore tried to address this gap in knowledge by studying one key example, a Form I rubisco that lacks the small subunit. All forms of rubisco are assembled from the basic functional building block of dimers (L_2_), where two large subunits (RbcL) are assembled head-to-tail. This is the smallest known catalytically active form of rubisco. Form I rubiscos, the most ecologically abundant form of the enzyme, are hexadecameric holoenzymes (L_8_S_8_) composed of four dimers with eight small subunits (RbcS) that cap both ends of the junction between adjacent dimers. The small subunit is unique to Form I rubiscos, so it has traditionally been thought that RbcS was integral to both Form I protein stability and its evolutionary history [29]. However, a novel clade of rubiscos (Form I’) lacking small subunits, a sister to Form I, has recently been discovered through metagenomic analyses, and a representative octameric rubisco (L_8_) was successfully purified and kinetically characterized [30]. Other, novel closely-related clades of L_8_ rubiscos (Forms I-ɑ and I’’) have also been recently discovered in a similar fashion [31]. Form I’ rubiscos likely diverged before the evolution of Cyanobacteria and the small subunit [30]; therefore, studying rubiscos from this clade presents a unique opportunity to study the effect of evolution on rubisco KIEs. We therefore measured in vitro the KIE of an L_8_S_8_ Form I rubisco from *Synechococcus elongatus* PCC 6301 in comparison to the KIE of an L_8_ Form I’ rubisco from *Candidatus* Promineofilum breve. We found the fractionation to be smaller in the L_8_ rubisco compared to the L_8_S_8_ rubisco (16.25 ± 1.36‰ vs. 22.42 ± 2.37‰, respectively). Our results imply that while the presence of a small subunit is not necessary for protein function, it may affect the KIE. Our findings may help provide insight into the function of the small subunit and allow more refined interpretation of carbon isotope data in environments, past and present, where Form I’ rubiscos may be unknowingly operating.

## 2. Materials and Methods

### 2.1. Delta Notation (δ^13^C)

Carbon isotope data were reported using delta notation (δ^13^C) in units of per mille (‰) where δ^13^C = [^13^R_sa_/^13^R_ref_ − 1]*1000, where the subscripts ‘sa’ and ‘ref’ denote sample and reference, respectively and ^13^R is the ratio of ^13^C/^12^C. All values in this study were reported relative to the Vienna Pee Dee Belemnite (VPDB) reference.

### 2.2. Rubisco Purification

The rubiscos used here were purified according to previous methodologies and had their kinetics characterized previously [30,32]. Briefly, 14xHis-bdSUMO-tagged *Candidatus* P. breve rubisco and untagged *S. elongatus* PCC 6301 rubisco were expressed in BL21 DE3 Star *E. coli* cultures. P. breve enzyme was prepared by conducting Ni-NTA affinity purification on clarified lysate, followed by subsequent purification by anion exchange chromatography and size exclusion chromatography. *Syn*6301 enzyme was prepared by subjecting clarified lysate to ammonium sulfate precipitation at the 30–40% cut, followed by subsequent purification by anion exchange chromatography and size exclusion chromatography. The enzyme was then stored on dry ice and the KIE assay performed within one week. UCSF ChimeraX (version 1.5) was used for visualization of protein models and preparation of manuscript figures [33,34].

### 2.3. Rubisco KIE Assay Summary

We used a substrate depletion method to measure the KIE of each rubisco as used previously in similar studies [25,35,36,37]. Briefly, this method relies on measuring the time-varying δ^13^C value of the CO_2_ pool as the reaction goes to completion instead of directly measuring the difference in δ^13^C between the initial CO_2_ and final 3PG pool. The KIE is then calculated from these data using a Rayleigh relationship, which considers the log-log transformation of the CO_2_ isotope data against the fraction of substrate consumed. Linear regression of these data can then be converted to a measure of the instantaneous isotope fractionation—the empirical measure of the isotope effect associated with rubisco carboxylation. With this formulation, larger KIEs correspond to steeper slopes in a Rayleigh plot. 

The assay mix we used is based on previous similar studies. In this set-up, inorganic carbon is supplied as HCO_3_^−^ which is converted to CO_2_ by a carbonic anhydrase (CA), typically derived from bovines. CO_2_ and RuBP is then catalyzed by rubisco to create 3PG. Therefore, our reaction mixture contains CA, rubisco, HCO_3_^−^, and RuBP to yield the full reaction, and additional reagents including: (i) MgCl_2_ (Sigma-Aldrich, St. Louis, MO, USA) to support correct rubisco active site metalation, (ii) bicine (Sigma-Aldrich, St. Louis, MO, USA) as a buffer, and (iii) dithiothreitol (DTT) (Sigma-Aldrich, St. Louis, MO, USA) to prevent rubisco oxidation and degradation [38].

In our experiment, instead of limiting CO_2_, we limited RuBP. In addition, *f* (the proportion of CO_2_ remaining) is typically known from an external measurement. Prior experiments have labored to constrain *f* by taking a separate aliquot of the assay to measure CO_2_ concentration directly [25,36]. In our experiment, we converted sampling time to *f* by fitting our data to the model y = a*EXP(−b*x) + c based on the fact that the δ^13^C of the reactant pool will increase during the reaction and then asymptote to a fixed value as the reaction ceases (i.e., no further carbon isotope discrimination can occur because rubisco can no longer pull from the CO_2_ pool as RuBP runs out). In essence, we are interested in the curvature of this line, similar to prior rubisco assays where the δ^13^C of the reaction vessel headspace was monitored continually on a membrane inlet mass spectrometer [35] instead of traditional methods where discrete aliquots are taken [25]. See below and Supplemental for further discussion. 

### 2.4. Assay Preparation and Execution

Prior to running the KIE assay, the activity of bovine erythrocytes CA (Sigma Aldrich; St. Louis, MO, USA C3934) was checked following manufacturer guidelines [39]. We found a value of 3368 W-A units/mg protein, which exceeded the product stated value of ≥2000 W-A units/mg protein, and so we proceeded to use this active CA enzyme prep in the KIE assay.

Glass sampling vials with septum tops (‘Exetainer,’ 12 mL, Labco, Lampeter, UK) were prepared. First, three external standards were prepared by weighing out Carrara marble standards (CIT_CM2013, δ^13^C = 2.0 ± 0.1‰) into individual exetainers. Standards were then sealed within each tube, purged with He gas for 5 min, and then acidified by needle injection with concentrated phosphoric acid (42% *v/v*) (Sigma-Aldrich, St. Louis, MO, USA). Then, three HCO_3_^−^ substrate exetainers were also sealed, purged with He gas, acidified by needle injection of phosphoric acid to convert HCO_3_^−^ to CO_2_, and placed in a 70 °C water bath for at least 20 min. Finally, 22 exetainer sampling vials were prepared for the rubiscos (12 for L_8_, 10 for L_8_S_8_). All sampling tubes were first sealed and purged with He gas for 5 min, and then injected with ~1 mL of anhydrous phosphoric acid (Sigma-Aldrich, St. Louis, MO, USA). The phosphoric acid both stops the reaction progress and converts all dissolved inorganic carbon species into CO_2_ for analysis.

Next, the reaction assay for each rubisco was prepared. First, a CA stock solution was made by dissolving bovine erythrocytes CA into DI water. Next, an RuBP stock solution was made by dissolving D-Ribulose 1,5-bisphosphate sodium salt hydrate (Sigma Aldrich; St. Louis, MO, USA R0878) in DI water. Then, one drop of concentrated hydrochloric acid (38% *v/v*) was added to 20 mL of autoclaved DI water while it was simultaneously stirred with a stir bar and vigorously bubbled with N_2_ gas for 10 min to remove any residual HCO_3_^−^ or CO_2_. Then, while N_2_ gas was blown over the surface of the solution to inhibit O_2_, reagents were added to create a final concentration of 100 mM bicine, 30 mM MgCl_2_, 1 mM dithiothreitol (DTT) (St. Louis, MO, USA), and 6.25 mM NaHCO_3_ (St. Louis, MO, USA). pH was adjusted to 8.5 with NaOH and HCl. CA from the CA stock was added, and then either the L_8_ or L_8_S_8_ rubisco was added to the solution. We used 0.996 mg of L_8_S_8_ and 1.18 mg of L_8_ rubisco. The solution was gently bubbled with N_2_ gas for 10 min while rubisco ‘activated.’ While the solution was bubbling, the syringes used for each rubisco assay were rinsed with ethanol and water. We used a separate 25 mL gas-tight syringe with a sample-locking needle for each rubisco (Ref #86326, Model 1025 SL SYR, Hamilton Company, Reno, NV, USA). 

RuBP was then added to each reaction assay and mixed through pipetting and swirling. This entire solution was then quickly transferred to a gas-tight syringe. The first time point (t = 0 min) was taken as quickly as possible after transfer. To sample, ~1 mL of the reaction assay was injected into the pre-prepared sampling exetainer containing phosphoric acid. Each assay was sampled 10–12 times over 390 min. 

A control was run in a separate experiment, where all the assay components were mixed together with the exception of a rubisco enzyme. The δ^13^C of the measured headspace did not change appreciably during this time period, with δ^13^C = −0.42 ± 0.03‰ at 0 min and δ^13^C = −0.55 ± 0.03‰ at 277 min. The absolute values of these measurements reflect the δ^13^C of the substrate used on that experimental day and cannot be related to the data shown here.

### 2.5. Isotopic Measurement

The δ^13^C of CO_2_ in the headspace of each exetainer was measured on a Delta-V Advantage with Gas Bench and Costech elemental analyzer (Thermo Scientific, Waltham, MA, USA) at Caltech. Before measuring samples, two tests were performed to ensure the instrument was functioning normally: (i) An ‘on/off’ test with an internal CO_2_ standard for instrument sensitivity and to establish a baseline intensity at a ‘zero’ CO_2_ concentration, and (ii) a linearity test where the concentration of CO_2_ was increased linearly within the designated sensitivity range of the instrument to ensure that a linear increase in CO_2_ concentration corresponds to a linear increase in electrical signal on the collector cups. We measured the concentration of ^12^CO_2_ at mass 44, and ^13^CO_2_ at mass 45. The instrument was also tuned to ensure that each mass was measured at the center of its mass peak. 

The headspace of each sample and standard was measured 10 times (10 analytical replicates), with an internal CO_2_ reference run before and after each suite of 10 analytical replicates. Data were visually inspected to ensure the sample was being measured within the correct sensitivity range of the instrument (i.e., of similar intensity and pressure as the internal CO_2_ reference). The ‘raw’ δ^13^C values were then corrected relative to VPDB using the three external standards. Assay results can be seen in Appendix A.

### 2.6. Calculation of KIE

We first pre-processed the data by assessing which data points to fit. We expected the δ^13^C of CO_2_ to increase following an exponential curve that eventually reaches an asymptote, but the last few data points start to decrease in δ^13^C. This may be due to a variety of reasons, including: (1) Ambient CO_2_ contaminating the exetainer containers as they are left out after the reaction; (2) re-equilibration of the aqueous and gaseous inorganic carbon pools; or (3) instrument error upon needle sampling of exetainer vial. Because exponential curves are linear in a log-log space, we therefore log-transformed the data points then systematically fit a linear regression through varying sets of data and calculated the resulting error (adjusted R^2^ value). The adjusted R^2^ value consistently decreased after data point 9 for L_8_, and after data point 8 for L_8_S_8_ (Appendix A). Therefore, we proceeded to use data points 1–9 for L_8_ and 1–8 for L_8_S_8_.

We then converted time to *f*, the fraction of the inorganic C pool remaining. Since RuBP was the limiting substrate, we could calculate the moles of CO_2_ consumed if we assume: (i) A 1:1 ratio of RuBP to CO_2_ was utilized by Rubisco, and (ii) full consumption of the RuBP pool. For each rubisco assay, 125 μmol of RuBP and 9.84 μmol of NaHCO_3_ were added. Therefore, 7.87% of the initial CO_2_ pool was consumed, or *F* = 0.9213. We then assume that *f* = 1 at *t* = 0, and *f* = 0.9213 at the upper bound of the fit. A general model of y = a*EXP(−b*x) + c was applied to the data, though with carbon isotope data in the ^13^R format instead of the δ^13^C format because ^13^R values can be manipulated arithmetically while δ^13^C values cannot [40]. The model was then fitted using the non-linear least squares function (call: *nls*(); R Statistical Software (v4.1.0; R Core Team 2021, [41])). Model outputs are shown in Appendix A. 

Time was then converted to *f* using the equation:$$f=1-\left(\frac{R_{i}-R_{0}}{R_{upper}-R_{0}}\times \left(1-F\right)\right)$$
where *R_0_* is the first measured ^13^*R* value in each set of data, *R_upper_* is the fitted parameter *c* from the model and *F* = 0.9213, which is calculated from the amount of RuBP added to the assay. 

Next, a correction was done to account for the C isotope fractionation between CO_2_ and HCO_3_^−^ at equilibrium, where CO_2_ is ~8‰ lighter (more negative δ^13^C value) than HCO_3_^−^ [42]. We followed the correction outlined in [25] where the adjustment is applied before linear regression in a Rayleigh plot:$$R/R_{0}\;adj.=\frac{{\left(fR/R_{0}\right)}^{C}}{f}$$
where *C* = (1.009 + 10^(pK − pH))/(1 + 10^(pK − pH)). The pK is that of carbonic acid, for which we used a value of 6.4 [43]. The pH of the L_8_S_8_ assay was 8.49, and the pH of the L_8_ assay was 8.52. 

Finally, a Rayleigh plot was made for each rubisco plotting ln(^13^*R*/^13^*R*_0_)_*adj*._*1000 vs. −ln(*f*) (Appendix A). The best fit slope, *D*, was calculated using a linear regression (call: *lm*(); R Statistical Software (v4.1.0; R Core Team 2021, [41])). *D* was then converted to Δ, the KIE, using the equation Δ = *D*/(1 − *D*/1000) [25]. Doing so, we found the KIE of the L_8_S_8_ rubisco to be 22.42 ± 2.37, and 16.25 ± 1.36 for the L_8_ rubisco. Results are shown in Table 1.

## 3. Results

### L_8_ Rubisco Has Smaller KIE than Its L_8_S_8_ Counterpart

The KIE of the L_8_ rubisco is ≈5‰ less than that of the L_8_S_8_ rubisco (16.25 ± 1.36‰ vs. 22.42 ± 2.37‰, respectively; Table 1). We note that there is variation among KIE measurements of similar or the same strains. Prior measurements which we compare our data against (Figure 1, Appendix A) are bacterial (Form II, Form I’) or Cyanobacterial (Form I) rubisco measurements, where a pure enzyme, substrate-depletion assay such as ours was performed on well-characterized strains where rubisco was obtained through expression and subsequent purification from *E. coli*. We also included one Form II measurement from a *Riftia pachyptila* symbiont, *Candidatus* Endoriftia Persephone [44], where rubisco was purified from the host trophosome because at the time of the measurement the symbiont could not be cultured separately from the host, though a complete genome has recently been published [45]. Therefore, we did not include measurements where a non-native bacterial rubisco was expressed by another organism in vivo and KIE calculated by extrapolating ratios of intracellular to extracellular CO_2_ [46], nor measurements from plants or the *Solemya velum* symbiont because it is not a member of the Cyanobacteria [36]. It has been proposed and measured that rubisco KIEs vary with pH, temperature, and metal ion concentrations [47,48], yet other studies contradict this claim [49] and have instead proposed that much of the variation in the literature reflects experimental uncertainties rather than intrinsic variations in KIE [16]. This study and [50] measured an L_8_S_8_ rubisco KIE from *Synechococcus elongatus* PCC6301 and 7942, respectively (identical RbcL and RbcS sequences) in similar assay conditions but found values that are similar but do not overlap in uncertainty, supporting the conclusion that variations in reported KIE values are due to experimental uncertainty rather than intrinsic enzymatic variations. However, the KIEs presented in Figure 1 were measured in assays that span a range of pH, temperature, and MgCl_2_ concentrations (Appendix A), notably with increasing MgCl_2_ concentration corresponding with increasing KIEs measured in the Form II rubisco by [25]. Because of the lack of repeated, rigorous measurements of multiple rubisco KIEs across variations relevant parameters (i.e., pH, temperature, metalation), it is difficult to conclude what is causing the variation in KIE values across studies. Therefore, we can only conclude that the L_8_ rubisco KIE is less (by roughly 5‰) than its L_8_S_8_ counterpart measured in this study, and less than the range of L_8_S_8_ rubiscos measured from previous studies.

Similarly, compared to prior Form II (L_2_) rubisco KIE measurements, the Form I’ (L_8_) rubisco may fractionate less. Compared to Form I KIEs, there is wider variation in previously measured Form II KIEs, with the Form I’ rubisco measured here overlapping in value with one Form II rubisco within uncertainty [53]. We note that all the Form II data presented here are largely from one species, *Rhodospirillum rubrum*, though the specific strain is not reported for all studies. Therefore, the variations may reflect experimental uncertainty with the exception of the measurement in [25], where MgCl_2_ concentration was changed. Therefore, we are not confident concluding either way if the L_8_ KIE is less than the L_2_ KIE or not. 

## 4. Discussion

### 4.1. Presence or Absence of RbcS External to Active Site May Influence KIE

Rubisco KIEs have also been used to support conclusions gleaned from non-isotopic kinetic parameters, both to better understand the reaction mechanism and to offer complementary data to traditional measurements, but our results belie an easy interpretation within that existing framework. The dominant theory in this field posits that rubisco specificity is positively correlated with the CO_2_ KIE because of an observed increase in carbon isotope fractionation, but not oxygen isotope fractionation, with specificity [15,25]. This argument originates from studies of deuterium (D) isotope effects on enzymatic reaction rates, which have been traditionally performed because deuterium displays a much larger (and easier to measure) KIE due to the large relative mass difference between D and its major isotope, H, in comparison to other rare isotopes such as ^13^C vs. ^12^C or ^15^N vs. ^14^N [65]. These foundational experiments have led to the conclusion that the isotope effect is determined at the rate-limiting step at the transition state, and small asymmetries in the transition state caused by transition state structure will cause small variations in the isotope effect [65,66]. Applied to rubisco, [15] proposed that the inherent difficulty in binding a ‘featureless’ CO_2_ vs. O_2_ molecule has caused natural selection in the transition state, where rubiscos that maximize the structural difference in transition states for carboxylation vs. oxygenation are able to be more specific. That then causes a trade-off where greater resemblance to the final carboxyketone product causes the product to also be tightly bound, leading to a higher S_C/O_ correlating with a lower V_C_, but also a prediction that the intrinsic KIE for CO_2_ addition (but not O_2_ addition) should increase as the transition state becomes more product like, i.e., higher-specificity rubiscos should have higher KIEs, which is indeed what the data at the time supported [15]. This has also led to the conclusion that rubisco is actually perfectly optimized for the time and places where it is found today, precluding any opportunity to use rubisco engineering to achieve increased biomass yields [15].

However, new CO_2_ KIE measurements that do not show a correlation with specificity are empirically questioning this conclusion (Figure 1B). Prior studies [37] have pointed out that the spread in KIE data, particularly at high specificity, cannot easily be described by a simple inverse relationship or linear regression. Indeed, our Form I’ measurement lies below the original regression line (dashed line in Figure 1B) proposed in [15]; its KIE is effectively too low given what one would predict via its specificity. However, although an increasing spread in CO_2_ KIE becomes apparent as more rubiscos are measured, they cannot directly address the dominant theory because of the general dearth of O_2_ KIE measurements. In addition, specificity is typically not reported in the same study with KIE (see notes in Appendix A), so some of the spread in Figure 1B may be due to uncertainties in the true specificity for the given rubisco measured. Therefore, additional paired measurements of CO_2_ and O_2_ KIEs with specificity are necessary before a new theory relating isotopic and non-isotopic kinetics can be proposed; more data are needed to decide between potential theories.

In addition, this transition state optimization theory is based on the assumption that it is the active site (which binds the intermediary carboxylation or oxygenation product) that concurrently affects both specificity and KIE, so the naïve assumption is that the absence or presence of the small subunit, which does *not* contain the active site, should not affect KIE. Unexpectedly, the L_8_ rubisco fractionates roughly 5‰ less than that of the L_8_S_8_ rubisco (16.25 ± 1.36‰ vs. 22.42 ± 2.37‰, respectively). The specificity of the L_8_ rubisco is indeed less than that of the L_8_S_8_ (36.1 ± 0.9 vs. 56.1 ± 1.3, respectively, [30]) but this may be a coincidence because that prediction is based on a theory reliant on rubisco’s active site which the small subunit does not directly impact. Our comparative study suggests the tantalizing hypothesis that the small subunit increases rubisco KIEs. However, Form I’ has only recently been discovered [30] and only a limited number of sequences exist. Future work consisting of dual CO_2_ and O_2_ KIE measurements of other novel Form I’ rubiscos compared to Form I rubiscos, across a range of assay parameters, will be needed for a more robust comparative study. Potentially, comparative studies of extant L_8_ vs. L_8_S_8_ rubiscos could be complemented with experiments using ancestral rubiscos demonstrated to not require RbcS–RbcL interactions [51] that would allow one to effectively strip the small subunit from an L_8_S_8_ rubisco and measure its effect on the KIE. Similarly, pairings of one RbcL sequence with various RbcS sequences of tobacco rubiscos [67] would allow one to test how various small subunits affect the KIE in Form I (L_8_S_8_) rubiscos. Alternately, it has been shown that mutations distal from the active site affecting oligomerization can affect enzyme kinetics, which is somewhat analogous to losing RbcS in that does not directly interact with the active site. KIE measurements from such rubiscos may also help shed help shed light on the relationship between RbcS, specificity, and KIE [68]. Therefore, it remains an open question what structural and biochemical aspects of rubisco may also affect KIEs in addition to active site and transition state theory mechanisms. 

### 4.2. Supports Prior Work Positing That Rubisco KIEs Vary across Phylogeny in the Modern Day and across Time

Our work supports previous work showing that the rubisco KIE varies across phylogeny in the modern day, though with the caveats that few unique rubiscos have been measured, there is variation across experiments, and the vast majority of measurements are from Form I rubiscos (Figure 1B, and see [13,37] for recent compilation across phylogeny). A smaller KIE measured from one novel Form I’ rubisco, in comparison to the bacterial Form I rubiscos, supports this observation, though more measurements across the Form I’ clade are needed to quantify any potential in-clade variation.

In addition, if we view L_8_ as an evolutionary ‘missing link’ between the evolution of L_2_ and L_8_S_8_ rubiscos, this measurement supports the idea that rubisco KIE may have varied across evolutionary time. Prior work has explored this question by measuring the KIE of a putative Precambrian ancestral Form IB rubisco reconstructed using a combination of phylogenetic and molecular biology techniques [54]; that study found the ancestral rubisco to fractionate less than its modern counterpart (17.23 ± 0.61‰ vs. 25.18 ± 0.31‰, respectively) [50]. Interestingly, the Form I’ and putative ancestral Form IB rubisco have similar, lower KIE values (16.25 ± 1.36‰ vs. 17.23 ± 0.61‰, respectively) compared to most modern Form I rubiscos (roughly 20-30‰; for recent review see [13]). This supports prior predictions that the KIE should have varied over geologic time in response to changing pCO_2_, though that prediction was based on an assumption of inverse correlation between specificity (selected for by changing CO_2_/O_2_ levels) and KIE [15]. This implies that the KIE of ancestral rubiscos may have been lower than modern rubiscos today, though this is a tentative hypothesis that, by necessity, relies on ancestral enzyme reconstruction and comparative biology techniques instead of direct measurements of ‘true’ ancestral enzymes. 

Finally, it is hypothesized that the small subunit may have evolved in response to rising atmospheric oxygen levels roughly 2.4 billion years ago because the high V_O_ stabilization that RbcS offers allows simultaneous exploration of RbcS and RbcL protein space [30]. Therefore, understanding the KIE of Form I’ rubiscos may allow us to better understand changes in rubisco biochemistry that may have accompanied evolutionary changes and facilitate better tracking of carbon mass flux at key times in Earth’s evolutionary history. 

## Figures and Tables

**Figure 1 biomolecules-13-00596-f001:**
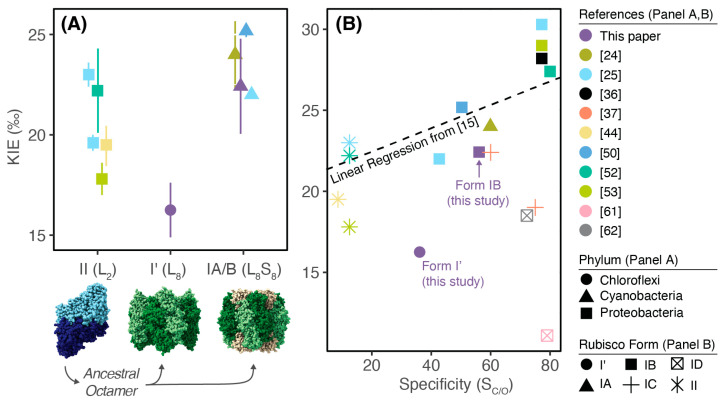
Form I’ rubisco fractionates less than both Form II and Form I rubiscos, and cannot be explained by prior model relating specificity and KIE. (**A**) KIE (‰) for relevant bacterial Form II (L_2_), Form I’ (L_8_), and Cyanobacterial Form IA/B (L_8_S_8_) rubiscos with representative rubisco structures below; Protein Data Bank (PDB) codes from left to right: 5RUB, 6URA, 1RBL. Hypothesized evolutionary pathway is shown in black arrows, showing that ancestral dimers (L_2_) likely evolved to a common ancestral octamer (L_8_) [51] that then speciated into either Form I’ (L_8_) or Form I (L_8_S_8_) rubiscos [30]. Rubisco phylum is shown as shapes and references are shown in colors. Form II KIEs are from *Rhodospirillum rubrum* or *Candidatus* Endoriftia persephone [25,44,52,53], Form I’ measurement is from *Candidatus* Promineofilum breve (this study), all Form IB rubiscos are from *Synechococcus elongatus* PCC6301 or 7942 (identical RbcL and RbcS sequence) [25,50] and this study, and Form IA KIE is from *Prochlorococcus marinus* MIT9313 [24]. Error is reported as 95% confidence intervals for [24]; as standard deviation for this study and [50,52,53]; as standard error for [25]. See Appendix A for literature values used, notes on variation between measurements, and rationale for which data was included and excluded. For recent compilation of all measured rubisco KIEs, see [13]. (**B**) Compilation of additional KIE and specificity values in Form IC and ID rubiscos [25,30,36,37,52,54,55,56,57,58,59,60,61,62,63], in addition to data shown in Figure 1A. Forms shown in shapes, references shown in the same colors as in Panel A. See Appendix A for compilation of data used. Dotted line indicates original linear regression from [15]. Figure was prepared with the assistance of the *ggplot2* package (v.3.3.66; [64]).

**Table 1 biomolecules-13-00596-t001:** KIE and non-isotopic kinetic measurements from L_8_ vs. L_8_S_8_ rubiscos.

Strain	Rubisco	KIE (‰)	V_C_ (s^−1^)	K_C_ (μM)	S_C/O_	V_O_ (s^−1^)	K_O_ (μM)
*Synechococcus elongatus* PCC6301	L_8_S_8_	22.42 ± 2.37	14.3 ± 0.71	235 ± 20.0	56.1 ± 1.3	1.10	983 ± 81
*Candidatus* Promineofilum breve	L_8_	16.25 ± 1.36	2.23 ± 0.04	22.2 ± 9.7	36.1 ± 0.9	1.11	401 ± 115

KIEs were measured in this study using the substrate depletion method [25,35,36,37]; see Methods for more detail. Non-isotopic kinetic measurements are from [30]. V_C_ and V_O_ indicate maximum carboxylation and oxygenation rates under substrate-saturated conditions, respectively; K_C_ and K_O_ are Michaelis constants for the carboxylation and oxygenation reactions, respectively; S_C/O_ indicates specificity, a unitless measure of the relative preference for CO_2_ over O_2_ and is calculated as S_C/O_ = (V_C_/K_C_)/(V_O_/K_O_). Uncertainties on non-isotopic kinetics reflect mean ± s.e.m. from multiple experiments; see [30] for more detail. Error on KIEs reflect mean ± s.d. from model fitting uncertainty from one experiment; see Methods and Appendix A for more detail.

## Data Availability

All data used in this study are presented in the Appendix A.

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
