# Peer review of "A Bacterial Form I’ Rubisco Has a Smaller Carbon Isotope Fractionation than Its Form I Counterpart"

_biomolecules, 2023, doi:10.3390/biom13040596_

Round 1
Reviewer 1 Report
The photosynthetic carboxylase enzyme Rubisco links the inorganic and organic aspects of Earths carbon cycle via the fixation of carbon dioxide (CO2) into sugar. Rubisco discriminates between naturally occurring isotopes (carbon-12 and carbon-13) of its substrate CO2 with a kinetic preference for carbon-12 termed the kinetic isotope effect (KIE). The strength of this preference differs along the phylogeny of Rubisco enzymes spanning billions of years of evolutionary distance leading them to have different KIE values. KIE values are useful for tracking mass flux through modern and ancient environments. Therefore, understanding the KIE values of uncharacterized Rubisco improves the accuracy of geological dating of material derived from organic matter.
The authors Wang et al attempt to shed some light on the KIE diversity of Rubisco by performing an initial measurement for the recently identified Form I’ Rubisco isoform which likely represents enzymes that existed in autotrophs billions of years ago. The authors are experienced with this new Rubisco, having been the first to discover it only several years ago.
Form I’ Rubisco exhibits unique and surprising catalytic parameters that would have previously been thought of as impossible when considering its quaternary architecture. A determination of its KIE value is therefore highly interesting simply from a mechanistic perspective, but also adds to our understanding of how KIE by Rubisco may have changed throughout evolution. The authors are well positioned to make these measurements and do so carefully. I would recommend this manuscript for publishing in biomolecules.
There are some points that must be addressed or clarified by the authors prior to this.
1. Reference 46 is unacceptable. You cannot reference unpublished works. I suspect/hope that this is another paper the lead author currently has in submission or is perhaps in press. The unaccounted-for-reference is used several times through the paper and features critically in figures. I cannot find any trace of it online or in preprints. At worst this could be considered fraudulence and therefore must be explained by the authors and corrected or removed with an explanation on the impact for the manuscript’s conclusions.
1. The authors seem to be oddly selective for certain KIE values from the literature, and their selection criteria don’t make much sense. I can accept that there are important differences between KIE measured from Rubisco activity in vitro or in vivo as in vivo processes can have additional enzymes or diffusion effects that influence the ratio of CO2 isotopes present around the active site. However, in some cases, such as for the excluded data of Ref 48, the Rubisco was extracted from the natural host. Why should this matter compared to Rubisco extracted from bacteria, as was performed by the authors in this study? The authors should include comparative KIE values in their analysis for all measurements performed in vitro using Rubisco enzyme extracts.
Minor points:
1. Define the acronym Rubisco in the first instance.
2. Line 15-16: “that the enzyme complex assembles without small subunits” this is confusing and makes it sounds like all Form I Rubisco can assemble without small subunits, or that Rubisco without small subunits are somehow novel. Neither are true. Consider clarifying for the reader.
3. In table S2 P. breve specificity reference is listed as ‘this paper’ where it should be listed as Banda et al. 2020. Similarly one of the references for PCC6301 is listed as ‘this paper’ (there was no Rubisco specificity measurements performed in this paper!), and in another it references the work of the non-existent Reference 46.
Reviewer 2 Report
The authors performed a very thorough and detailed analysis of carbon isotope discrimination (KIE) of a more recently discovered cyanobacterial Form I’ rubisco that lacks small subunits (L8) in comparison to the canonical cyanobacterial Form IB rubisco (L8S8), and present convincingly that L8 rubisco has a lower KIE. This is a very exciting finding as it is generally thought that the small subunits are not affecting catalytic properties as the active site is on the large subunits, and that KIE scale with rubisco specificity (sc/o) which in turn is thought to depend on the large subunits alone.
In their comparative analysis of KIEs between Form I’, IA/B and Form II rubisco and correlations with sc/o, value obtained in other studies were included whereby the authors selected carefully studies with similar experimental conditions.
The manuscript is very well written and very easy to follow which makes reading enjoyable. The intention and significance of the study are clearly explained and put into context. Understanding KIE and the evolutionary trajectory of rubisco structure will aid interpretation of earth’s biological paleohistory. This study taps into a novel aspect to consider in the application of KIE data such as metabolic modelling and reconstruction of ancient environmental scenarios.
One aspect which could perhaps be discussed further is how widely the Form I’ is distributed amongst cyanobacteria. One begins to wonder what proportion of total biomass may carry the Form I’ KIE signature and whether this proportion has changed through geological times.
I am not certain I fully agree with the view that bioengineering a better rubisco would be reasonable only if rubisco properties of today were an evolutionary accident (line 61 to 65). Perhaps one could add the idea that rubisco is likely optimal for photosynthetic organisms as they have slowly evolved with changing environments on evolutionary time scales. However, with accelerated anthropogenic climate change the need to adapt requires likely more than evolutionary speed and humankind exerts a different ‘selection pressure’ asking for high yielding plants useful to humans but not necessarily in the best interest for adaptation and survival of the species. Hence bioengineering could be viewed as the attempt to accelerate evolution, except that we do not have enough understanding of structure-function of rubiscos at the protein and molecular level to make informed and sophisticated changes to the enzyme.
At the technical level, I would suggest for the future to compare KIE obtained with the method described here to other measurement techniques for instantaneous KIE to verify the robustness of the observed differences.
Minor comments:
There seems to be a typo in line 31, where it should read Bassham not Basshom.
In the Discussion it seems sections 4.2. and 4.3 should be combined under a new descriptive headline.
Reviewer 3 Report
Form I Rubisco is the most abundant protein on earth, catalyzing global CO2-fixation in photosynthesis. Rubisco is infamous for its low catalytic efficacy and dual functions in photosynthesis and photorespiration. This paper studied the kinetics of Form I’ Rubisco, which consists of octameric Rubisco large subunits without small subunits. They found that kinetic isotope effect (KIE) is smaller in Form I’ Rubisco than that in Form I Rubisco from Synechococcus elongatus PCC6301. The authors concluded that small subunits might affect Rubisco’s KIE.
The paper is overall well written and the results are convincing. But the conclusion is worthy to debate. The authors drew the conclusion after comparison of Form I and Form I’ Rubisco from two organisms. It is desirable if there are more data to compare Rubisco from more organisms and more kinetic measurement.
Round 2
Reviewer 3 Report
The manuscript is improved. It is suitable to publish.